# Is the Fluorine in Molecules Dispersive? Is Molecular Electrostatic Potential a Valid Property to Explore Fluorine-Centered Non-Covalent Interactions?

**DOI:** 10.3390/molecules24030379

**Published:** 2019-01-22

**Authors:** Arpita Varadwaj, Helder M. Marques, Pradeep R. Varadwaj

**Affiliations:** 1Department of Chemical System Engineering, School of Engineering, The University of Tokyo 7-3-1, Hongo, Bunkyo-ku 113-8656, Japan; varadwaj.arpita@gmail.com; 2National Institute of Advanced Industrial Science and Technology, 1 Chome-1-1 Umezono, Tsukuba, Ibaraki Prefecture, Ibaraki 305-8560, Japan; 3Molecular Sciences Institute, School of Chemistry, University of the Witwatersrand, Johannesburg 2050, South Africa; helder.marques@wits.ac.za

**Keywords:** halogen-centered noncovalent interaction, weak F···F fluorine bonding, electrostatic potential, dispersion, QTAIM characterization, first-principles studies

## Abstract

Can two sites of positive electrostatic potential localized on the outer surfaces of two halogen atoms (and especially fluorine) in different molecular domains attract each other to form a non-covalent engagement? The answer, perhaps counterintuitive, is *yes* as shown here using the electronic structures and binding energies of the interactions for a series of 22 binary complexes formed between identical or different atomic domains in similar or related halogen-substituted molecules containing fluorine. These were obtained using various computational approaches, including density functional and *ab initio* first-principles theories with M06-2X, RHF, MP2 and CCSD(T). The physical chemistry of non-covalent bonding interactions in these complexes was explored using both Quantum Theory of Atoms in Molecules and Symmetry Adapted Perturbation Theories. The surface reactivity of the 17 monomers was examined using the Molecular Electrostatic Surface Potential approach. We have demonstrated *inter alia* that the dispersion term, the significance of which is not always appreciated, which emerges either from an energy decomposition analysis, or from a correlated calculation, plays a structure-determining role, although other contributions arising from electrostatic, exchange-repulsion and polarization effects are also important. The 0.0010 a.u. isodensity envelope, often used for mapping the electrostatic potential is found to provide incorrect information about the complete nature of the surface reactive sites on some of the isolated monomers, and can lead to a misinterpretation of the results obtained.

## 1. Introduction

The halogens in Group 17 of the periodic table [1], when covalently bound in molecules, can accept and donate halogen bonds; the latter phenomenon has been called halogen bonding, or simply X-bonding, or XBs [2,3,4,5,6,7]. Much effort has been made to demonstrate the role of halogen bonding in the design of engineered crystals [7], polymers, in biological systems, and molecular drugs [8]. Several hundred reports have appeared that help provide an understanding of factors influencing halogen bonding in complex molecular systems (for example, References [2,3,4,5,6,7,8,9,10,11,12,13,14,15,16,17,18,19,20,21,22,23,24,25,26]). It has been concluded that the heavier halogens (X=Cl, Br, I) in molecules are readily polarizable. They are dispersive and can therefore form halogen bonds with negative sites [2,3,4].

Halogen bonding interactions have been demonstrated to be electrostatically driven and highly directional [3]. However, Stone has challenged this, and has put forward the view that the high directionality of halogen bonding is the consequence of exchange repulsion, and depending on the extent of the attraction between interacting sites, a specific type of interaction dominates [9]. Others have shown that the σ-hole centered halogen bond model (see below) cannot explain the details of the non-linear nature of an intermolecular interaction in CH_3_OCZCl^+^···X complexes (Z=H, F, Cl; X=F^−^, Cl^−^, Br^−^) [27]; there have been a number of discussions in the literature that support this view [28,29,30,31,32,33,34,35,36]. 

A σ-hole on atom X is generally found along the outer extension on the R–X bond axis, where R is the remainder part of the molecule [2,3,4,5,6]. The electrostatic potential associated with this has been attributed to a deficiency in electron density. Traditionally, this deficiency was assigned a positive electrostatic potential, for which a bound halogen can donate a halogen bond. Yet while a σ-hole initially was thought to be positive by nature, it need not necessarily be so. It can be either positive, or negative, or even electrically neutral depending on the polarizing or electron-withdrawing capacity of the fragment R [10,11,12,14,37,38,39,40]; that the σ-hole can be negative has been pointed out in several recent studies [12,14,37,38]. Similarly, there have been suggestions that a σ-hole on a bonded atom in molecules can be induced through polarization by the electric field of the interacting molecule [41]; this has been criticized a number of times [11,33,42]. Although some firmly hold to this belief [41], others argue this is not so as a σ-hole is an inherent property of bound atoms in molecules that cannot be induced [42]. The placement of a positive/negative test charge at a distance from an atom in a molecule actually makes the entire system cationic/anionic, and the computed electrostatic potential will undoubtedly be positive/negative everywhere on the surface of the resulting molecular species [33]; this is certainly not due to induction. 

The propensity of covalently bound halogen to form a halogen bond generally follows the order R–Cl < R–Br < R–I, in line with the polarizability of the halogen derivative (Cl < Br < I) [2,3,4]. However, when a F atom shows a positive σ-hole on a given molecule and is involved in halogen bonding with a negative site, then (also taking astatine into account [43]) the stability ordering would become R–F < R–Cl < R–Br < R–I < R–At. 

There are a number of studies [44,45,46] that show there is no σ-hole on F along C–F bond extension in fluorobenzene (C_6_F_6_); this is misleading. It is indeed there, but negative. It is invisible because the lone-pair regions around the lateral sites dominate. Why a σ-hole is not found when X=F, as well as in some other instances, e.g., CH_3_Cl, was previously explained as the result of the higher electronegativity of fluorine/chlorine that gives it a disproportionately large share of the σ bonding electrons, which helps to neutralize the σ-hole [5]. This explanation is not true, since there is a negative σ-hole on the extension of the C–F bond in CH_3_F, which is graphically invisible [38,40]. The misconception that the σ-hole on F in C_6_F_6_ and CH_3_F is absent, or that it cannot be a negative, probably arose because the concept of the σ-hole [5] initially was not fully understood. As Bayse pointed out [30], even though halogen bonding interactions are often discussed in terms of an area of positive electrostatic potential on the halogen center along the outer portion of the bond axis, several workers [28,29,30,31,32,33,34,35,36] have noted a lack of completeness in this model. 

Several reports claimed that the lightest element of the halogen family, fluorine, does not participate in halogen bonding in molecules [5,47,48]. For instance, the crystallographic survey of Auffinger et al. identified a small set of short F···O contacts [48]. They did not consider these contacts for further study because the fluorine was assumed to be highly electronegative in character. Although a number of studies have been reported in later years that demonstrate the ability of fluorine to participate in σ-hole centered halogen bonding [10,21,22,24,39,40], Neaton, in his perspective article in 2017 [49], reintroduced the idea that “the σ-hole is more marked for Br and I than for Cl; for F, the lightest halogen, with its tightly bound electrons, ostensibly no σ-hole and no-halogen bonding would be formed”. This is a view that was first advanced 10 years ago [5,6]. There is a perhaps related view, aired some time ago, that fluorine in molecules rarely forms hydrogen bonds [50]; however, there are many reports that show that fluorine in molecules can be an excellent hydrogen bond acceptor [44,51,52]. Nonetheless, the supposed inability of the fluorine to form halogen bonds has arisen from the notion that it is spherical; that it is not polarizable; that it cannot be dispersive [17]; that it is small and has a high electronegativity; and so on [18,19]. These also led to a view that the electrons of fluorine in molecules are not easily redirected to the σ-bond and there is no significant depopulation of the *p_z_*-atomic orbital opposite to this bond [5,18,19]. Consequently, the fluorine shows only a minimal σ-hole on its surface. Clearly, these views purportedly do not allow fluorine to display a σ-hole, and may explain why fluorine is generally considered not to be a good halogen bond donor [18,19].

Despite the views summarized above, it is evident that the fluorine atom in many molecules is negative. It readily serves as a site of attraction for electropositive hydrogen atoms on another molecule, leading to the formation of hydrogen bonding interactions [51]. When a fluorine atom is covalently bonded to a strongly electron-withdrawing atom or group, in rare cases it shows a positive σ-hole on its outer electrostatic surface on the extension of the R–F covalent bond (for example, as in F_2_ and CF_4_) [7,10]. This positive site serves as a fluorine bond donor and has the ability to form fluorine-centered halogen bonding interactions with negative sites [10,25].

Since the fluorine in some molecules along the outer portion of its axial site is positive, its equatorial sites in many cases are negative. This, however, is not always the case. For instance, the lateral and axial sites of the fluorine in NCF (cf. Figure 1c) and NCCCF (cf. Figure 1e) are both positive so that both sites can serve as halogen bond donors. 

The lateral negative sites on the fluorine atom can serve as an acceptor of a halogen bond, as well as an acceptor of a hydrogen bond, depending on the extent of its negative character. Interestingly, however, Kawai et al. [23] have suggested that the negatively charged fluorine in molecules also has the potential to attract the entirely negative fluorine atom in another molecule, leading to the formation of ordered 2D supramolecular architectures on some crystalline metal surfaces. We [12,14] and others [13] have demonstrated that entirely negative fluorine atoms in fluorinated molecules (such as C_6_F_6_) can attract each other to form binary complexes in the gas phase, which explains the non-covalent feature observed in crystals in the solid state. These results point to the potential ability of fluorine to participate in a versatile range of non-covalent interactions, where it can act both as a donor and as an acceptor of electron density, as do other halogens in molecules [2,4,5]. Several studies (for example, References [21,22,23,24]) have made an effort to remove the widespread misconception regarding the inability of fluorine to participate in halogen bonding. However, given that research in this area is in its infancy compared to that on the heavier halogens, there is a clear need to enhance our understanding of the chemical reactivity of fluorine.

These brief introductory remarks deal not only with recent advances in our current understanding of halogen-centered non-covalent interactions formed by fluorine, but also the misconceptions prevalent in the field. It is clear that the non-covalent chemistry of fluorine is not yet fully understood, and the misconceptions present entry points for follow-up studies that will permit us to arrive at a better understanding of its bonding features. 

Some have suggested that the role of fluorine in crystal packing cannot be predicted [20]. However, there are others who have observed that the replacement of hydrogen by fluorine in molecules can lead to dramatic changes in the solid state packing behavior as a consequence of its weak interactions with other sites. This has implications for optoelectronic applications [8,53,54,55]. Berger and coworkers [43] have shown that fluorine wears many hats in a protein environment, and that there are numerous entirely untapped aspects of the C–F bond that are waiting to be explored in such an environment, including charge–dipole interactions and metal–cation coordination. The peptide community is surely ready to usher in a new era that will focus on the uptake, toxicity, and metabolism of organofluorine building blocks. 

In addition to the ability of fluorine in molecules to form hetero-molecular complexes [56], it would be interesting to see the way the fluorine atoms in molecules that are accompanied with sites of similar (e.g., positive) electrostatic potential forms homo-molecular complexes. To shed some light on this, the current study presents another aspect of fluorine chemistry in some molecules by exploring a series of 17 molecules (Scheme 1); these molecules are partially or fully fluorinated, or halogenated. These also represent prototypical model systems for exploring the reactivity of fluorine surfaces locally in molecules. 

It is appreciated that there can be many binary and higher order complexes formed if combinations of two or more monomers of all the 17 monomers chosen in this study are considered. Several of them may result when the positive site on the halogen in a monomer interacts with the negative site on the halogen on the other monomer, leading to the formation of a halogen bonded complex. As mentioned above, this specific arrangement between positive and negative sites in diverse molecules has been thoroughly examined, and much is known about the underlying chemistry of the non-covalent interactions involved. 

Among other aims, this study endeavors to answer the fundamental questions posed in the title. We aim to address whether the positive site on fluorine atoms in the selected fluorine-substituted molecules has the ability to attract the positive site on another halogenated molecule, governing the formation of the 22 bimolecular complexes examined. If it does, is the electrostatic energy that arises from Coulombic interaction between the interacting species of similar (or different) electrostatic potential sufficient to explain the attraction between them? Is the fluorine in the selected monomers polarizable, can it be dispersive? We note further that lump-hole type non-covalent interactions have been used to explain the intermolecular bonding interactions between halogen atoms in several intermolecular complexes [57,58]. Is the same lump-hole concept applicable to the bimolecular systems examined in this work to successfully explain the F···F, F···Cl and F···Br attractions between the atomic regions centered on the interacting halogen atoms that have identical (positive) electrostatic potentials on the monomers? 

To address the questions raised above, we focused on a specific type of arrangement between these 17 monomers in a 1:1 stoichiometry. For reasons discussed in the following section, similar results were also obtained with ab initio Møller–Plesset second-order perturbation theory, MP2. The binding energies for the complexes evaluated with the two approaches were compared with that obtained from coupled cluster CCSD(T) theory for which the low-level M06-2X geometries were used. The Quantum Theory of Atoms in Molecules (QTAIM) [59,60,61] approach was employed on top of these for the exploration of the chemical bonding topologies in the binary complexes. The Molecular Electrostatic Surface Potential (MESP) model [62] was used for the exploration of the nature of various reactive regions on the electrostatic surfaces of the 17 halogen-substituted monomers. The Symmetry Adapted Perturbation Theory (SAPT) [63,64] approach was employed to analyze the dissected energy components responsible for the interaction energies of the binary complexes examined. We have selected all these approaches because of their demonstrated importance in revealing the chemistry of non-covalent interactions in chemical systems. 

## 2. Computational Details

The 17 halogen-substituted molecules utilized for the construction of the 22 1:1 binary complexes examined in this study are shown in Scheme 1. These monomers are either fluorinated, chlorinated, or brominated, or mixed halogenated. They were drawn using the Gaussview 05 software package [65]. 

Both the M06-2X DFT functional and double-ζ basis set were previously shown to be adequate to obtain correct values of electrostatic potential [2,3,4,5,6,7,10,11,12,66,67]. However, from our previous experience [10,11,12], we found that the double-ζ basis set is not always reasonable to provide insight into the detailed nature of the surface reactivity of monomers; we therefore used the 6-311++G(2d,2p) Gaussian basis set of triple-ζ valence quality. We chose [M06-2X/6-311++G(2d,2p)] to fully optimize the geometries of all the monomers, which were then used to evaluate the 0.0010 a.u. (electrons bohr^‒3^) isodensity envelope mapped potential; selection of this isodensity envelope was based on previous recommendations described elsewhere [2,3,4,5,6,7,38]. For reasons also discussed elsewhere [10,11,12,68,69], the 0.002 a.u. isodensity envelope was also used to see what extra information it provides that was not provided by the 0.0010 a.u. isodensity envelope. Details of the underlying mathematical equations for calculating the MESP have been discussed many times previously [2,3,4,5,6,7,38], and are therefore not repeated here. The local surface maxima and minima (*V*_s,max_ and *V*_s,min_, respectively) of potential were used to characterize the nature of various site specific electron density localizations on the halogen atoms of the 17 monomers. 

One of our interests was to examine how strongly the fluorine atom in a given monomer binds the interacting atom in the partner molecule. Are the binding energies of the resulting complex systems assessed using a standard (low-level) computational method, such as the DFT-M06-2X method, comparable with those obtained from MP2? Or, if the results are within the intrinsic error of the computational method, are these methods therefore insufficient to extract reliable conclusions on the qualitative nature of the bonding interactions involved? Is it always essential to perform high level calculations to gain insight into the nature and presence of chemical bonding in chemical systems, even though it is often difficult to model larger systems (such as drugs) with high level first-principles theory? To answer these questions, the electronic structure calculations for all dimers were performed with the global hybrid M06-2X DFT functional, together with the 6-311++G(2d,2p) and 6-311++G(2df,2pd) basis sets. The choice of DFT functional was due to its reliability when studying non-covalent interactions as has been discussed (see [67,70,71] for example). In particular, Politzer and co-workers [4,38,72,73] have demonstrated on several occasions that this functional is reliable for the modeling of halogen-centered non-covalent interactions. However, given that the studied systems include heavier halogen derivatives, these might require treatment with extensive electron-electron correlation effects; we therefore used MP2. 

Each of the 22 dimer geometries were constructed using the Gaussview 05 software package [65]. Each of them was then optimized with [M06-2X/6-311++G(2d,2d)] and [M06-2X/6-311++G(2df,2pd)] without geometry constraints, as was done for the monomers, (i.e., none of the atoms in either of the geometries was fixed during the calculations of the first derivatives of energy with respect to the nuclear coordinates). Hessian second derivative calculations were performed for all cases to examine the nature of the optimized geometries; default thresholds for forces and displacements were used for first and second derivative calculations of energy with respect to the atom fixed nuclear coordinates. Tight convergence and a more accurate numerical integration (ultrafine) grid, was invoked. The Gaussian 09 program package was employed for the calculations [74]. In all instances, the resulting (optimized) geometries, shown in Figure 2, were slightly perturbed, and were close to the initial configurations. While Gaussian 09 has several algorithms for Self-Consistent-Field (SCF) calculations, the default SCF procedure with DIIS is quite fast and works well for most systems, with no damping or Fermi broadening, and was used. We have also performed similar calculations (no geometry constraints) with MP2/6-311++G(2df,2pd) to obtain the geometries and binding energies of all the dimers, which were compared with those obtained with M06-2X. For both DFT and MP2 methods, the density = current protocol was used. The counterpoise procedure of Boys and Bernardi [75] as implemented in Gaussian 09 was used for correcting the binding energies of all the complexes; the latter were obtained using the standard “supermolecule method” discussed by Pople [76]. 

Many previous studies have reported binding energies of complexes calculated using a high-level theory on a low-level geometry (often done in benchmarking DFT functionals). Although this procedure has been used to assess the quality of density functionals, it does not give very accurate energies. Should these be treated with skepticism? Moreover, our interest was to see whether any additional insight into the nature of dispersion can be obtained with correlated methods compared to those evaluated using M06-2X and MP2. For this, CCSD(T)/aug-cc-pVTZ single point calculations were performed on the M06-2X/6-311++G(2df,2pd) optimized geometries of 21 binary complexes. For the (C_4_F_6_)_2_ dimer in Figure 2d, a CCSD(T) calculation was not possible because of its size (20 atoms with 12 F atoms) and the limitation of available computational resources. 

According to the QTAIM of Bader [59,61,77], there exists a mapping between the molecular structure and the charge density *ρ*(**r**) observed from a diffraction experiment or calculated from electronic structure theory. From this picture, which is concordant with the Hellmann–Feynman theorem [78,79], the *ρ*(**r**) governs all interactions in the system and emerges as the conceptual bridge between structure and properties of a system, irrespective of whether the system in a dimer, a cluster, or an extended system in the solid state [80]. Two descriptors of QTAIM, bond path and their associated (3,−1) bond critical points (bcps), provide an indication about the presence of a bonding interaction; this is true regardless of whether this concerns intra- or inter-molecular interactions. Lecomte et al. [80] have argued that individual atom–atom pair interactions evaluated with QTAIM are structure determining, and thus are responsible for the entire structure of the chemical system being examined. According to them, when bond paths between atomic domains are observed, and where classical models prohibit bonding, these cases may indicate a failure of these models to elucidate unusual bonding situations for which they were never designed. The underlying theoretical details of the methodology has been discussed before [59,60,77], with some impressive contributions from the group of Tognetti and Joubert [81,82,83,84,85,86]; the potential implication of the topological approach has recently been discussed [87]. Taking these views into account, QTAIM calculations were performed with M06-2X/6-311++G(2d,2p) to examine what insight these offer in terms of the bond path and critical point topologies for all the 22 binary molecular complexes examined. Although such calculations at a higher level of theory are likely to produce different values for these properties, the conclusions arrived at on the reliability of the intermolecular interactions will remain unchanged. Both the Multiwfn [88] and AIMAll [89] packages were used. 

For reasons discussed previously, as in [90,91,92], SAPT-based energy decomposition analysis was performed with a truncated aug-cc-pVDZ basis set, called jun-cc-pVDZ, using PSI4 [63]. The underlying reason for selecting this approach was to provide insight into the approximate origin of the fluorine-centered non-covalent interactions between the interacting monomers leading to the formation of the complexes examined. The fundamental detail of the SAPT procedure is briefly outlined in the Results and Discussion section below. Because the calculated binding energies of the binary complexes are significantly smaller than the magnitude of electrostatic potentials on the surfaces of the halogen atoms in the isolated monomers (see below), the former and the latter properties are given in kJ mol^‒1^ and kcal mol^‒1^, respectively.

## 3. Results and Discussion 

### 3.1. The Nature of the Electrostatic Potential on the Surfaces of the Halogen Atoms in 17 Monomers

Is the 0.0010 a.u. isodensity envelope mapped potential, as often suggested for the evaluation of positive and negative regions on atomic surfaces in molecules, adequate to extract the realistic nature of the surface reactivity of all the 17 monomers chosen? The MESP model is used for inferring the reactive sites on isolated monomers [38] that can potentially serve as sites for electrophilic or nucleophilic attack. Figure 1 illustrates the 0.0010 a.u. isodensity mapped potentials on the electrostatic surfaces of all the 17 monomers, shown in Scheme 1. As can be seen, the σ-hole on the fluorine in most of these monomers is found to be positive. They are colored blue/green and occur along the outermost extension(s) of the R–F covalent bond. The fluorine in FCN and FCCCN is entirely positive and their σ-holes are largely delocalized. Provided they are involved in halogen bond with a negative site, these may or may not be directional. This is consistent with the findings of Stenlid et al. [93] in that the s-orbital type σ-hole is spread all over the H atom in molecules, which explains why this atom presents with weak directionality in its hydrogen bonding. 

For FCOOH, FNO_2_ and SOF_2_, however, the σ-hole on the fluorine has a different character. It is negative along the outermost extensions of the C–F and N–F bonds in FCOOH and FNO_2_, and absent on the outer surface of the fluorine along the S–F bond(s) in SOF_2_.

The nature and strength of the σ-holes on the halogen atoms in monomers are elucidated by examining the sign and magnitude of the *V*_s,max_ along the R–X bond extensions. These are summarized in Table 1. Among the 17 monomer geometries analyzed, the outer surface of the bromine along the Cl–Br bond in ClF_2_Br has the largest *V*_s,max_ of +56.18 kcal mol^‒1^. This shows that the σ-hole on the bromine in this molecule is largely deficient of electron density, and has significant potential to form a halogen bond. By contrast, the σ-holes on the fluorine atoms along the outer C–F bonds in CF_4_ are weakly positive, with *V*_s,max_ ≈ +0.42 kcal mol^−1^. 

These results suggest that the areas of positive electrostatic potential associated with the σ-holes on the halogens in monomers have the ability to attract negative sites to form σ-hole centered halogen bonding interactions. This is consistent with the view of others, *viz*., the forces that govern the structures of molecular crystals are always Coulombic interactions between regions of positive and negative charge [37,38]. Since σ-hole regions on these halogen atoms in the chosen molecules are positive, they should not attract the positive site on another species [2,3,4,5,6,10,11,21,22,24,25,94]; if they do so it would be counter to the fundamental principle of Coulomb’s law of electrostatics [95].

Now the question arises: Are the predicted 0.0010 a.u. isodensity mapped surface maxima of electrostatic potential *V*_s,max_ (σ-hole strengths) on the halogen atoms in monomers correct? They may be correct as long as the strengths of the positive σ-holes on the halogen surfaces are reasonably strong. However, when these are weaker, misleading conclusions could be reached [10,11].

To provide proof of this hypothesis, let us consider the 0.0010 a.u. isodensity mapped *V*_s,max_ values associated with the σ-holes on the halogen atoms as summarized in Table 1. The *V*_s,max_ on F along the N–F bond in FNO_2_ is calculated to be –0.06 kcal mol^−1^. There is no σ-hole on the fluorine atoms along the Cl–F, Cl–F and S–F bond extensions in ClF_3_, ClF_2_Br and F_2_SO, respectively. One would thus interpret the latter result to mean that the σ-hole on the F atom in these systems is neutral, as was done in previous studies on other systems (see [49,96] for example). One might also conclude that there could be an anomeric effect that strengthens polar flattening and allows facile polarization to give a “neutral σ-hole”—as has been done for the molecular fragment –CF_3_ [97,98]—that acts as a hotspot for dispersion interactions. 

Following the findings in some previous studies [10,11,68,99], we have calculated the *V*_s,max_ on the 0.0020 a.u. isodensity envelope for all the monomers and the results are listed in Table 1 as well. We find that the σ-holes on the fluorine atoms along the outer extensions of the Cl–F, Cl–F and S–F bonds in the three molecules mentioned above (ClF_3_, ClF_2_Br and F_2_SO) are all negative and are neither absent nor neutral, as found on the 0.0010 a.u. isodensity envelopes. For instance, the *V*_s,max_ is –22.12 kcal mol^−1^ on the fluorine along the outermost extension of the Cl–F bond in ClF_2_Br. It is −13.75 and −3.50 kcal mol^−1^ on the fluorine atom along the extensions of the Cl–F and S–F bonds in ClF_3_ and F_2_SO, respectively. Similarly, the *V*_s,max_ associated with the σ-hole on the fluorine in FNO_2_ is now not negative as found on the 0.0010 a.u. isodensity mapped potential surface, but positive, with *V*_s,max_ = +1.69 kcal mol^−1^ (hence displaying a positive σ-hole on the surface of the fluorine). 

To further demonstrate the unreliability of the 0.0010 a.u. isodensity envelope, we chose the F_2_SO system as a prototype, and have included in the bottom of Table 1 the *V*_s,max_ on F computed on various isodensity envelopes for F_2_SO, with values starting from 0.0010 a.u. to 0.0055 a.u. As is apparent from the data, the surface of bonded fluorine along the S–F bond extensions does not enclose a σ-hole when the potential is mapped with 0.0010–0.0015 a.u. isodensity envelopes; *V*_s,max_ is absent on these envelopes. However, choosing isodensity envelopes > 0.0015 a.u. leads to the appearance of *V*_s,max_ on F atoms along the S–F bond extensions. The sign of the *V*_s,max_, which is associated with the F’s σ-hole, remains negative passing from 0.00165 a.u. to 0.0050 a.u. isodensity envelopes, and it becomes positive when the 0.0055 a.u. isodensity surface was used. These variations on the sole character of the F’s σ-hole are certainly not the effect of polarization. 

These results unequivocally indicate that the 0.0010 a.u. isodensity envelope is not always appropriate for the prediction of the nature of electrostatic potential on the surface of a chemical system, especially when the monomer contains poorly polarizable F atom(s). This is not very surprising given that the 0.0010 a.u. isodensity envelope on which to compute molecular electrostatic potential is arbitrary [100], and not robust. This means the choice of an appropriate isodensity envelope has a significant impact in revealing the actual (and complete) nature of the reactive profile of an atom in molecules. This is consistent with the observation of Ibrahim [99], and others [68], who have recommended the use of a >0.0015 a.u. electronic density envelope, since these have shown to meaningfully produce a positive region on the chlorine atom in chloromethane [11] that was often taken to be negative [42,96,101] or neutral [5]. This is not very surprising, as different basis sets provide a different sign and magnitude for *V_S,max_* [10,11,12,66]. One might argue that choosing a > 0.0015 a.u. envelope means moving closer to the nucleus of the atom in molecules, and one might then come up with a tiny electron density deficient surface. While this is so, its use assists in extricating the missing potential extrema on the 0.0010 a.u. mapped isodensity envelope (as shown above). We note further that the two contour values (0.0010 and 0.0020 a.u.) were recommended by Bader et al. [69] and are large enough to encompass ≈ 96% of a molecule’s electronic charge—the selection of either of these envelopes depends on the nature and size of the atomic constituents in a molecule; these determine their overall volumes, polarities and so on. This study has often been referred to validate the claim that the 0.0010 a.u. envelope of the electronic density is the ideal molecular surface on which to compute the electrostatic potential [38]. However, our results are in agreement with the recent suggestion of Bauzá and coworkers [68] that the 0.0020 a.u. envelope, which encloses >99% of the electron density, is ideal for mapping of the potential. 

### 3.2. The Binary Complexes

#### 3.2.1. Intermolecular Geometries 

The M06-2X level energy-minimized 1:1 intermolecular complexes are given in Figure 2. Two of the most important geometrical properties are illustrated for each complex. The first is the X···F (X=F, Cl, Br) intermolecular distance. Its value varies in the range of 2.7–3.2 Å. The second is the angle of approach, pertinent to the directionality of the intermolecular interaction (see below). 

The van der Waals radius of a fluorine atom is 1.46 Å [102]. Twice this radius, 2.92 Å, is in some cases less, and in others more, than the F···F intermolecular distances calculated for the binary complexes illustrated in Figure 2. For instance, for the F_2_NF···FNF_2_, FOOCF···FCOOF and FCCF···FCN complexes the F···F intermolecular distance is smaller than 2.92 Å, but for all other F···F bonded complexes it is greater. Similarly, for ClF_2_Br···FClF_2_, the van der Waals radii sum, 3.32 Å, for Br (1.86 Å) and F (1.46 Å) atoms is larger than the Br···F intermolecular distance of 2.965 Å. Also, for ClF_2_Cl···FClF_2_ and Cl_2_···F_2_, the corresponding radii sum, 3.28 Å, for Cl (1.82 Å) and F atoms is greater than the Cl···F intermolecular distances of 2.971 and 3.198 Å, respectively. These results demonstrate that while the IUPAC recommendation of “*less than the sum of the van der Waals radii*” for identifying a halogen bond may be reasonable for medium-to-strongly bound complexes [103], application of this criterion on its own is certainly not suitable for recognizing weakly bound halogen-centered non-covalent interactions in molecular complexes. 

To shed some light on this, we imposed upper and lower limits for the F···F distance of 1.60 and 3.50 Å, respectively, and a residual *R* factor < 0.075, while searching the Cambridge Structure Database (CSD) [104] for the –CF_3_···F–C– fragments in crystals; we excluded powder diffraction structures, structures with errors, and structures whose coordinates are not deposited in the CSD. This resulted in a total of 116,475 –CF_3_···F–C– fragments in 13,727 crystal structures. The most intriguing aspect of this result was that the largest frequency (6998 occurrences) occurs in these crystals for F···F distances vary between 3.15 and 3.20 Å. The great majority of these distances lie between 2.90 and 3.30 Å, a range which is either close to, or much greater than, twice the van der Waals radius of the F atom (1.46 Å), indicating that these weak interactions have profound implication in crystal growth and design. 

We note further that whether the geometric criterion discussed above is appropriate to study non-covalent interactions has been debated several times recently, although there are some who believe this criterion is robust. The use of this criterion was cautioned against some 10 years ago [105], but it is still widely used, especially in the crystallography literature. Some of us [10,12,14,40,106], and Politzer and coworkers [37,38], have examined the appropriateness of assigning “close contacts” in crystal lattices using this criterion. For example, it has been argued that close contacts, which are defined as interatomic separations less than the sum of the respective van der Waals radii, are commonly invoked to identify attractive non-bonded interactions in crystal lattices. While this is often effective, especially for reasonably medium to strong interactions, it can also be misleading because (a) there is significant uncertainty associated with the van der Waals radii, and (b) it may not be valid to attribute the interactions solely to specific pairs of atoms. The interactions within crystal lattices are Coulombic, and the strongest positive and/or negative regions do not always correspond to the positions of atoms; they are sometimes located between atoms. 

#### 3.2.2. Directionality 

Directionality is one of the major features of halogen bonding and other σ-hole interactions, including hydrogen bonding [7], pnictogen bonding [107], and tetrel bonding [108] and so on. The bound halogen in molecules are known to form two types of contacts. These are termed type-II and type-I contacts. The former is directional and the latter is not. This is so as the angle of approach of the electrophile (the halogen with a positive σ-hole) associated with the former is often found to be linear or quasi-linear (Scheme 2), whereas for type-I interactions the angle associated with the halogen atoms is always non-linear. It has been shown that highly directional halogen bonds are stronger than those that deviate significantly from 180° [7,109]. High directionality, together with the involvement between a positive and a negative site, is the primary requirement for the formation of a type-II halogen bonding topology, while such a requirement is not necessary for the formation of type-I interactions [110,111].

For the binary complexes illustrated in Figure 2, the halogen···halogen intermolecular interactions are evident between the interacting halogen atoms. For the formation of the Y···X–R non-covalent interactions in these complexes, there is no requirement for negative and positive sites to be involved in the attractive engagement because the interacting sites on the halogen atoms are both positive in the monomer species. This shows that the definition of type-II halogen bonding illustrated in Scheme 2 is not applicable to the intermolecular interactions found for complexes in Figure 2. Also, the Y···F–R interactions in the complexes of Figure 2 (except (NF_3_)_2_) cannot be called halogen···halogen type-I interactions [110,111] because the angles of interaction (θ = ∠Y···F–R) for these dimers of Figure 2 lie in the range 170° < θ < 180°; for type-I halogen bonding the angle is generally 90° < θ < 150° [110,111].

Scheme 3 introduces an analogous concept, which we call *σ-**σ centered type-II halogen-centered non-covalent bonding*. This terminology is applicable to systems formed between two atomic sites of similar or dissimilar electrostatic potential in interacting monomer molecules in which the interacting sites conceive σ-holes that are either both positive or both negative. 

The *σ-**σ centered type-II halogen-centered non-covalent bonding* pattern is evident in the binary complexes, shown in Figure 2, except (NF_3_)_2_, in which, the ∠Y′···X′–R′ varies between 170° and 180°. The intermolecular interactions in some of these complexes may not be regarded as of the lump-hole type [57,58], since the two halogen atoms in the isolated interacting monomer molecules attracting each other are equipotential and the ∠Y′···X′–R′ is 180°. (In a lump-hole interaction, one of the sites is relatively electron density deficient and the other electron density rich). 

It has been said that type-II halogen···halogen contacts are halogen bonds [112]. This notion is in sharp disagreement with the type-II halogen···halogen contacts in the dimers of Figure 2; a majority of these cannot be called halogen bonds [112]. Thus, all type-II halogen···halogen contacts cannot always be regarded as halogen bonds; for the formation of a halogen bond the primary requirement is that there must be a positive site on the halogen that must interact attractively with a negative site. This is evidently not the case for the halogen···halogen interactions found in the dimers of Figure 2. 

For the (NF_3_)_2_ dimer geometry illustrated in Figure 2u, ∠F···F−N = 116.2°. This shows the presence of a type-I halogen···halogen bonding topology [110,111]. Its formation is due to the attraction between the negative equilateral sites localized on the fluorine atoms in NF_3_, which could be driven by dispersion. 

For the (SOF_2_)_2_ and (FCOOH)_2_ dimers (Figure 2k,q), as well as a few others in Figure 2, the ∠Y′···X′–R′ angle deviates slightly from 180°. These are examples of dimer systems that do not involve secondary interaction between the interacting monomers, yet the intermolecular interaction is quasi-linear. In these, the regions associated with the two negative σ-holes on the fluorine atoms in the two different molecules are involved in an attractive engagement, providing stability to the resulting complexes. The deviation of the angle of approach from linearity in these might be an indication of the presence of a lump-hole interaction. This has explained the stability of a nearly slipped parallel (HBr)_2_ dimer [57] in which the Br···Br intermolecular interaction was the result of an attractive interaction of a lump in the Br atom of one HBr molecule with the hole on Br in the other HBr molecule. 

Because halogen bonds are more directional than hydrogen bonds, it has been suggested that halogen bonding is not merely a weak van der Waals type interaction [113]. Rather, halogen bonding is best viewed as including contributions from electrostatics, as well as the involvement of *n* → σ* charge transfer interactions, where *n* and σ* are the lone-pair and anti-bonding orbitals, respectively [113]. While this may be true when they are formed between positive and negative sites, we show below that this is not always the case and that halogen-centered σ-hole interactions can be weak and of van der Waals type. We also show that halogen-centered σ-hole non-covalent interactions cannot be fully described by Coulombic (and polarization) interactions, especially when the interacting partners are weakly bound. 

Huber and co-workers have suggested that that the synergy between charge-transfer interactions and Pauli repulsion are the driving forces for the directionality, while electrostatic contributions are more favorable in the less-stable, perpendicular orientation [32]. Riley et al., using high-level SAPT analysis, suggested that, as in the case of neutral halogen bonds, exchange forces are important contributors to the directionality of charged halogen bonds. However, it is also found that induction effects, which contribute little to the stability and directionality of neutral halogen bonds, play a large role in the directionality of halogen bonds involving charged species [114]. Others suggested directionality of the halogen bond is caused by electrostatic and exchange-repulsion energies [115]. Stone claimed the directional nature of the halogen bonded interaction is mainly driven by the exchange part of the interaction energy, and not just electrostatics, and this may change when the interaction becomes weaker. In such a case, other interactions, including dispersion, play a definitive role [9]. Clearly there is no consensus. This suggests that the origin of directionality in halogen bonding is yet to be fully explored, since the explanations offered vary from one system to another; they appear to depend on the magnitude of the interacting energy components on a system-by-system basis that are collectively responsible for the overall stability. It is difficult to say which force actually matters on what specific system without careful evaluation of the component energies. This is because the stability of any interaction and its directionality, and hence the complex as a whole, in our view, is determined through the delicate balance between all types of forces acting on the nuclei of atoms. Clearly, the net interaction energy arising from the attractive components must be such that they supersede the repulsive ones to result in a bound state. 

#### 3.2.3. QTAIM Characterization of Bonding Interactions 

The QTAIM molecular graphs of the 22 energy-minimized dimer geometries are shown in Figure 3. As noted in the Computational details section, the combination of the (3,−1) bcps and the bond paths determines the molecular graph. The (3,−1) bcp is a saddle point of charge density *ρ*(**r**) [61], and is a measure of bond strength [116]. Tognetti et al. have provided convincing arguments to show that bond paths are valuable indicators of bonding interactions and have provided explanations for the occasions where they disappear [81,82,83,84,85,86].

QTAIM provides an indication about all types of X···F intermolecular interactions between the atomic basins that are presumably bonded to each other in the 22 intermolecular complexes. Each molecular graph features one X···F intermolecular bonding interaction (except for (NF_3_)_2_, Figure 3u), identified on the basis of the characteristics of *ρ*_b_, the charge density at the bcp, and it Laplacian, ∇^2^*ρ*_b_. For instance, *ρ*_b_ at the X···F bcp is found to be very small for each dimer. The maximum value of *ρ*_b_ is 0.0080 a.u. for the Br···F interaction in F_2_ClF···BrClF_2_ (Figure 3j). Similarly, the ∇^2^*ρ*_b_ at the F···F, Cl···F and Br···F bcps are positive and small (Figure 3). These two fundamental signatures, together with the presence of a bond path, are presumably necessary [117] to characterize the X···F short contact distances in the 22 dimers as closed-shell type. This accords not only with IUPAC footnote feature F9 recommended for hydrogen bonding [118], but also with an IUPAC-recommended characteristic for halogen bonding [103]. 

For the (NF_3_)_2_ dimer there are three bcps between the F atoms of the two interacting monomers. Both *ρ*_b_ and ∇^2^*ρ*_b_ for the F···F interactions are small, and with ∇^2^*ρ*_b_ > 0. The three equivalent intermolecular distances of separation in this complex are 3.193 Å, and ∠F···F–N are 116.2°. As indicated already above, the latter suggests the presence of a type-I bonding pattern, and the QTAIM characteristic indicates the presence of closed-shell interactions between the fluorine atoms. One cannot conclude that the intermolecular interactions in this complex are σ-hole type as there is no σ-hole on the perfluoro ammonia molecule on the bonded side of the N atom along the outer extension of the C_3_ axis. Clearly, the concept of σ-hole theory cannot explain the intermolecular bonding interaction in this complex. 

The topology of intermolecular bonding in the (NF_3_)_2_ dimer is analogous to what has been reported previously for the binary complex formed between NH_3_ and CF_4_ [119]. In this complex, there were three N···F bond paths between lateral negative sites on the three F atoms of the –CF_3_ fragment in CF_4_ and the negative N in NH_3_. To avoid complexity, the F–CF_3_···NH_3_ molecular graph presented in that study did not include a detailed topology of various bond- and ring-critical points and their connectivity. What was shown was possible bonding interactions between atomic basins, which are probably not sufficient to describe the detailed topology of interactions involved. This is because the QTAIM analysis did not reveal the presence of any bonding topology between the C and N atoms in this complex, which can be regarded as a tetrel bond. This could be expected, since the C atom has a positive σ-hole along the F–C bond extension in CF_4_ and was facing the N atom of NH_3_. The missing bond path and critical point topologies between the C and N atoms are a clear indication that this might be a weakly bound interaction, and that QTAIM has a tendency to underestimate such interactions. One can thus readily conclude that neither MESP nor QTAIM alone can reveal the nature of detailed bonding features in the system described above (*viz*. F–CF_3_···NH_3_). However, in this specific case, as well as for (NF_3_)_2_, a combined analysis is seemingly beneficial to elucidate the presence of possible bonding interactions between the nuclei of atoms that are bonded with each other. 

From the results above, we conclude that the MESP approach is useful for recognizing the areas of positive and negative potential on the electrostatic surface of an isolated molecular domain. It provides insight into the anisotropic nature of covalently bound main group elements (including halogen) in molecules. The same insight can be obtained from 2D plots of ∇^2^*ρ* as regions of charge depletion and concentration [59,60,61,81,82,83,84,85,86].

While it is generally recognized that the MESP approach is useful to infer whether a particular atomic site on a molecular species can attract a site of opposite electrostatic potential on another molecular species [2,3,4,5,6,7], it is certainly not useful to elucidate what happens when the two sites on the isolated interacting monomers both have positive (or both negative) electrostatic potentials. In such a case, it is misleading to claim that any acceptor-donor interaction is driven by just Coulombic forces that have an *r*^−2^ distance dependence acting between negative and positive sites. Not all acceptor-donor interactions observed between the monomers in Figure 2 emerge from the attraction between negative and positive sites. Such a conclusion would have been arrived at if these interactions were viewed only on the basis of the electrostatic potentials on the surfaces of atoms in the isolated monomers. 

We note that it is difficult to provide direct proof whether there are positive σ-holes on the halogen atoms on the monomers in the equilibrium geometries of the dimers. To evaluate this, one might map the potentials for monomers when these are isolated, and then again when they are complexed. However, the electrostatic potential and the electron density on the interacting halogens are likely to be different in the two cases, accompanied by electronic structure changes upon dimer formation. Unfortunately, there is no current computational protocol available for predicting the electrostatic potential on covalently bound halogens that are already interacting non-covalently; this would be required to provide insight into the extent to which these were triggered on the formation of the equilibrium non-covalently bonded interaction. This failure is because the monomers available as fragments in the equilibrium geometry of a complex system have already overlapped (at least partially!) in this geometry. The scope of the MESP approach is limited to evaluating the potential on the interacting atomic surfaces of the monomers that have already overlapped. This point has been made previously by others [55]: “[A]n important point to keep in mind is that once a non-covalent interaction A···B, either intramolecular or intermolecular, has occurred, the regions of positive and negative potential on A and B that were the driving force for it will not be visible in the final A···B complex. They will have been at least partially neutralized by the interaction.” Notwithstanding this, σ-hole interactions have been judged primarily based on the nature of reactive sites on the isolated monomers, that is, the formation of σ-hole interactions in complexes has been rationalized solely based on the positive σ-hole computed on the electrostatic surface of an on atom in an isolated monomer and the negative site on another isolated monomer. This does not assist in determining, quantitatively, the role played by polarization as the energy associated with this cannot be quantified with the MESP model, even though its underlying concept has often been invoked in many reports to argue that the σ-hole interaction is a consequence of polarization. This matter has already created some controversy with terms, such as “failure of σ-hole theory” [29,42,120] being used. We will address this elsewhere.

Despite its limitations, QTAIM is indeed an important tool for the exploration of intermolecular interactions involving halogens. Its great advantage is that it enables an evaluation of the nature of the localization or delocalization of electron density in both isolated and interacting species. It provides insight into the nature of charge density in the bonding region. This makes it a more useful methodology than the MESP approach for exploring the possible existence of non-covalent interactions in a wide variety of situations, including the σ-hole···σ-hole driven molecular dimers examined in this study. Similarly, the advantage of the molecular orbital approach over MESP has recently been demonstrated by Angarov and Kozuch [28], and is yet another competing approach for analyzing non-covalent interactions. 

#### 3.2.4. Stability of Complexes: Quantification of Interaction Energies 

We examined the energy stability of the complexes illustrated in Figure 2. The stabilization energy (or binding energy) Δ*E* for each was computed with M06-2X/6-311++G(2df,2pd) using the relationship Δ*E*(dimer) = *E*(dimer) − *E*(monomer 1) − *E*(monomer 2), where *E* is the total electronic energy of each individual species. We have used a relatively large basis set for the calculation of Δ*E* because the dimeric complexes examined in this study are expected to be weakly bound, which was inferred based on the “less than the sum of the van der Waals radii” criterion discussed above.

The basis set superposition error energies, Δ*E*^corr^, were calculated at the same level using the counterpoise procedure of Boys and Bernardi [75]. The computed Δ*E* and Δ*E*^corr^ values are summarized in Table 2. 

As expected, the computed Δ*E* values for all the dimers are small. The values also indicate that the binary complexes illustrated in Figure 2 are not just of the van der Waals type. Some of them can be regarded as weakly bound. This classification is in accord with Desiraju [121], and our previous suggestion [106,122] that intermolecular complexes with bond energies around −1.0 kJ mol^−1^ can be treated as van der Waals complexes, whereas those above this value can be treated as weakly bound interactions, with stabilization energies between −1.0 and −17 kJ mol^−1^ [121,123,124]. For instance, halogen bond strength [45,99] is similar to that of the hydrogen bond, in the range 1–40 kcal mol^−1^ [5,6], or even larger; these latter have been classified as van der Waals, weak, medium, strong and ultra-strong [122]. 

The BSSE energy calculated with a relatively large basis set is not negligible, and is comparable in magnitude to Δ*E* or, in some cases, even larger (Table 2). The large BSSE is the cause for the relatively small Δ*E*^corr^ values of many of the complexes. The BSSE evaluated with DFT and first principles methods is known to be spurious [125,126,127], a feature which is also evident when a relatively large basis set, 6-311++G(2df,2pd), is used. 

The question arises whether the various dimer geometries (and their energetics) discussed above are an artifact of the M06-2X computations, or whether the same result is reproducible using other first principles methods, as well as that with other DFT functionals that have been designed to account for intermolecular interactions. While we did not investigate the reliability of the structure of the dimers and their stability by using other DFT or DFT-D functionals, we have repeated the same (geometry-optimization and Hessian) calculations with the MP2/6-311++G(2df,2pd) method. 

As expected, the MP2 method produced very similar results, *albeit* with shorter intermolecular bond distances, and a concomitant increase in the complex binding energies (see Table 3). The angles of approach for the intermolecular bond formation, as well as the nature of the eigenvalues of the Hessian second derivative matrix, evaluated with this method are comparable with those of the same complexes shown in Figure 2 (M06-2X). Of 22, eight binary complexes (e–f, h–i, q–r, t–u, Table 3) were found to have one or two imaginary frequencies, which represent first or second order saddle point structures. Geometries of this type have recently been the subject of a recent study [24]. This shows that the dimer structures illustrated in Figure 2 and their nature (minima or saddle points) are not an artifact of the M06-2X methodology. These results suggest the weakly bound complexes examined are real and their significance cannot be overlooked in the growth and design of the crystaline state of fluorine-containing compounds. 

#### 3.2.5. Energy Decomposition Analysis 

Several groups have reported that the SAPT approach [90,91,92,128] is very effective in obtaining physically meaningful insight into the nature of non-covalent interactions in molecular systems [129,130,131,132,133,134,135,136,137]. SAPT decomposes the interaction energy into four separate components: Electrostatics (*E*_els_), exchange (*E*_exc_), induction (*E*_ind_) and dispersion (*E*_dis_). It should be noted that there are various levels of truncation of SAPT that are implemented in the PSI4 code [63], including, for example, SAPT0, SAPT2, SAPT2+, SAPT2+(3) and SAPT2+3 [92]. Even though the computations are very expensive as the level of truncation increases, we have used the SAPT2+3 level of truncation in this study that accounts for the CCD dispersion. The net interaction energies, *E*(SAPT2+3), together with the RHF level energies, all evaluated within the SAPT framework, are listed in Table 2, and are compared with the uncorrected and counterpoise-corrected binding energies of the M06-2X/6-311++G(2df,2pd) and CCSD(T)/aug-cc-PVTZ//M06-2X/6-311++G(2df,2pd) methods. 

Nevertheless, the fluorine atoms in all the monomers were found to be negatively charged, as were the other halogen atoms in some of the other monomers examined in this study. Even so, these halogen atoms are involved in forming weakly bound intermolecular complexes (*vide infra*). What causes these negatively charged atoms in the halogen-substituted molecules to attract one another, even though their interaction energies are weak? 

As can be seen in Table 2, the sign of *E*_els_ for some binary complexes is negative, but positive for others. *E*_els_ < 0 for some complexes arises presumably because each monomer species in these complexes polarizes the other when the two are in close proximity, presumably creating image charge clouds of opposite polarities; hence the attraction between them produces a negative *E*_els_. This shows that the F···F interactions in these 14 complexes are at least partially stabilized by electrostatics. 

The electrostatic energy for some complexes of Table 2 is positive (*E*_els_ > 0). In particular, this is prominent for complexes, including F_2_···Cl_2_ (e), ClF_3_···ClF_3_ (g), ClF_3_···ClF_3_ (h), ClF_3_···Cl_2_F_2_ (i), ClF_3_···BrF_2_Cl (j), F_2_···FCN (n), NCF···FCN (o) and HCOOF···FCOOH (q). Intriguingly, this occurs especially when the sign and magnitude of the σ-holes on the halogen atoms of the monomers interacting with each other are appreciably both positive or negative (see *V_S,max_* values in Table 1). Clearly, *E*_els_ > 0 for these complexes might be a consequence of increased repulsion between the interacting sites of similar or dissimilar electrostatic potential. This shows that *E*_els_ plays the role of repulsion for the F···X (X=F, Cl, Br) interactions that are responsible for the geometries of these eight complexes. 

The induction energy, *E*_ind_, is calculated to be small for all dimers illustrated in Figure 2. This provides further evidence that the fluorine atoms in the monomers involved in the formation of these dimers could be polarizable, but weakly so. Irrespective of the nature of the sign of the electrostatic component, the data in Table 2 indicate that the interaction energy for each complex, obtained by summing over *E*_els_, *E*_exc_ and *E*_ind_, would be positive at the RHF level, since these three terms best describe the RHF level interaction energy. We have also energy minimized all the complexes with RHF/6-311++G(3df,2pd) and calculated their stabilization energies, as listed in Table 3. Except for ClF_3_···ClF_3_ (g), ClF_3_···BrF_2_Cl (j) and NCF···FCN (o), the uncorrected Δ*E* for the remaining dimers were found to be negative. However, when BSSE was taken into account at this level, the Δ*E*^corr^ for 18 complexes were found to be positive with RHF and that for the remaining four complexes (*viz*. SO_2_F_2_···F_2_SO_2_ (k), NCCCF···FCCCN (l), FCCF···FCN (p) and HCOOF···FCOOF (r) of Figure 2) were very weakly negative. This result provides further evidence that Coulombic interactions alone are insufficient and should not be used as the sole energy feature when evaluating the stability of dimeric systems. This result also runs counter to the proponents of σ-hole theory who have continually suggested that polarization, a part of dispersion, is an aspect of classical Coulombic forces, which provide a satisfactory and physically-based interpretation of non-covalent interactions [138]. However, our view is consistent with that of many others [28,33,34,139,140,141,142]. In particular, Thirman et al. [33] argue that it is impossible to predict halogen bond strength without accounting for charge-transfer and other contributing factors. 

An interesting feature of Table 2 is the magnitude of the dispersion energies. This is significantly larger than all other attractive terms for each binary system. Incorporation of this component actually makes the net binding energy large and negative, and is therefore largely responsible for the stabilization of the dimeric complexes examined. This could lead to the conclusion that the attraction between the regions of positive electrostatic potential on the interacting halogen atoms in the isolated monomers is primarily caused by the dispersive interaction between them that has an *r*^−6^ radial dependence. That like-charged sites can attract is consistent with a perspective already propounded by Lekner [95]. From the results of Table 2, it is also clear that contributions arising from polarization and dispersion interactions are not the same. Their existence has been known for a long time; treating each as a separate component sheds some light on the physical origin of a variety of intermolecular interactions. This isolation is also consistent with the nature of their radial dependences (*viz.* dispersion energy ∝ *r*^−6^) that have different significance in different contexts, which can be readily discriminated from a Coulombic interaction (*viz*. charge-charge, charge-dipole and dipole-dipole).

The SAPT energies, *E*(SAPT2+3), summarized in Table 2, are in reasonable agreement, at least qualitatively, with the BSSE corrected CCSD(T) binding energies, even though the latter are evaluated with a different basis set. This result shows that the SAPT approach is able to handle correlation effects, and so is an approach that is indeed useful for developing a fundamental understanding of weakly interacting systems, especially those that are weakly bonded and predominately of van der Waals type. The CCSD(T) results, as well as of the MP2, answer the questions posed in the Computational details section that concern whether the high-level calculations are necessary for the complexes examined of this study (see above). 

Echeverría et al. have recently reported a set of C–H···H–C interactions in some dimers of alkane that were “subtle but not faint” [143]. The 1:1 complex topology used was useful in understanding the effect of larger molecular skeletons on C–H···H–C interactions, where each such interaction responsible for stabilization of the dimer was by about 1 kJ mol^−1^. With the various 1:1, 1:2, 2:2, 1:3 and 3:3 complex topologies explored, the MP2 dissociation energies were calculated to be in the 0.52–1.75 kJ mol^−1^ range. Some of these authors in another study have demonstrated that many experimentally determined crystal structures deposited in the CSD [104] feature weak contacts. They focused their theoretical exploration on understanding the weak Hg···Hg contacts between the Hg atoms in some complexes [144]. The stabilization energies were found to vary between −0.84 and −11.92 kJ mol^−1^, obtained using various density functionals, together with MP2. In other studies, Echeverría and co-workers have examined M–H···H–C [145,146], C–H···H–C [147], and P–H···H–P [148] contacts in several crystallographically determined structures. The BSSE corrected stabilization for the CH_4_···H_3_Al complex was found to vary between −4.02 and −9.27 kJ mol^−1^ depending on the correlated level of theory used (MP2, DFT, DFT-D and CCSD(T)). The effect of the BSSE was severe, especially with MP2, but this is not very surprising because this method has the tendency to overestimate the BSSE. The range of binding energies reported for these non-covalent interactions is comparable to the energies computed for the X···F intermolecular interactions examined in this study. These results show the importance of weak interactions both in molecular and crystal design. Their importance should not be underestimated, as often occurs in many crystallographic studies.

The results of this study also support the recent work of Wagner and Schreiner [149] and of Rösel et al. [150,151] who have emphasized that the determination of inter- and intramolecular dispersion energy improves our ability to design sophisticated molecular structures and much better catalysts. The significance of London dispersion as an important element of structural stability has long been underestimated in molecular chemistry, probably because of the common notion that dispersion is weak. This may be true for pairs of interacting atoms, such as the ones discussed in this study. However, for increasingly larger structures, the overall dispersion contribution grows rapidly and can amount to tens of kcal mol^−1^ [150]. There are similar recent studies that demonstrate the importance of dispersion in molecular and crystal design [147,152,153,154,155,156,157,158,159].

## 4. Discussion

We have examined theoretically several fluorine-containing dimers and have shown that these can be formed by the attraction between the interacting halogen atoms where at least one of them is fluorine. We have shown the circumstances under which the fluorine can be polarizable and dispersive. Specifically, we have shown that the fluorine in a number of monomers has a positive region along the outermost extension of the R–F σ-bond, yet it can sustain a non-covalent bonding interaction with the positive σ-hole on a halogen in another molecule, leading to the formation of various binary complexes. Many of these displayed a *σ-**σ centered type-II halogen bonding* pattern; these are certainly not halogen bonds. This topology of bonding is not markedly different from the F···F directional interactions recently discussed by others and by some of us [12,14,23], where entirely negative/positive fluorine in the perfluoro molecules underwent a homo-molecular attractive bonding interaction with negative/positive Lewis bases. 

The above observations on the stability of binary complexes that emerged using the M06-2X functional were shown to be valid with MP2 and CCSD(T) calculations. Although the interaction energies for the complexes quantitatively differ on passing from one level of theory to the other, this is not unexpected given that the extent of correlation and exchange effects are differently incorporated in RHF, M06-2X, MP2, CCSD(T) and SAPT. One may not therefore assume that the binding energies reported are associated with the errors of the computational methods applied. 

In addition to the ability of fluorine in molecules to form non-covalent interactions discussed above, it is well known that fluorine in some molecules (for example, F_2_) has the ability to form different kinds of non-covalent interactions, such as F···H hydrogen bonding, X···F halogen- and σ-hole bonding. For the first, the lateral negative portion of fluorine confers on it the ability to act as a hydrogen acceptor, while for the second its axial portion serves as the site of halogen- and σ-hole bond donors (as in F_2_···NH_3_). The ability of fluorine to participate in a diverse range of attractive engagements justifies it being called the chameleon of non-covalent interactions. 

The results obtained also suggest that the dimers that arise as a consequence of the interaction between fluorine-containing molecules are not strongly bound. They may be regarded as either weakly bound or simply van der Waals complexes based on various classification schemes previously proposed [121,123,124]. While the 0.0010 a.u. isodensity envelope has often been invoked to study reactivity, we found that this envelope can give incomplete results, leading to misleading conclusions on the nature of the surface reactivity. 

The weakening of the fluorine-centered interaction is, in part, due to Coulombic repulsion, and, in part, due to exchange repulsion. These can be partially overcome by the attractive energetic component arising from induction, but this tends to play a minor role in determining the net stability of the complex. Dispersion has been shown to play a key role in determining the stability of these complexes. This conclusion, drawn with SAPT, is qualitatively consistent with the results emanating from MP2 and CCSD(T) calculations. 

We have also shown that type-II halogen···halogen interactions cannot always be regarded as halogen bonding, as previously suggested, because the same pattern of type-II bonding can be found in systems that do not involve attraction between positive halogen and a negative site. 

There is precedence to show that the *V_S,max_* on the halogen in an isolated monomer correlates with the binding energy of the halogen bonded complex, and can thus be regarded as a measure of bond strength [2,3,4]. While doing so, the importance of the electron density changes—expected when two systems come in the close proximity—associated with the interacting atoms upon the formation of the non-covalent interactions is neglected. For instance, complex formation leads to electronic structure (*viz*. bond length) changes associated with halogen donor and acceptor fragments, accompanied by changes in the vibrational characteristics [160]; these cannot be explained by the *V_S_*_,*max*_ on the halogen of the isolated monomer. This is because *V_S_*_,*max*_ cannot be calculated on the bonding region, since the atoms that are bonded non-covalently are already (at least partially) overlapped at equilibrium [55]. This limitation, however, does not prevent some from using the MESP model to provide arguments to explain what caused two interacting sites to attract one another. We did not observe any such linear scaling between the *V_S,max_* on the halogen in monomers involved in the formation of the F···X interaction and the binding energy for the series of dimers examined, a result in agreement with those of others [33,34,36,100,161,162]. 

Clearly, the intermolecular interactions between similar local polarities on halogen derivatives in molecules deserve further exploitation, since F···F interactions (and other similar interactions) are abundant in many crystals. These are arguably the least understood interactions, yet are likely to be at least in part responsible for the stability of many crystalline materials. 

This study has advanced our fundamental understanding of the role played by the other face of the fluorine, its dispersive ability, and highlights an important aspect of fluorine-centered non-covalent interactions. The importance of this interaction is evident in the recent study of Bauzá et al. [163], as well as that by us [164], in which it was reported that there are hundreds of crystallographic structures in the CSD that contain F···F interactions, and that are probably weakly bound. Their importance in the design of new crystalline materials is therefore unsurprising. 

## 5. Conclusions

We conclude that:The 0.0010 a.u. envelope on which to compute the electrostatic potential is arbitrary, and its use may mislead when attempting to explore the complete nature of the reactivity of the fluorine in some of the molecules examined. This is particularly shown for F along Cl–F, Cl–F and S–F bond extensions in ClF_3_, ClF_2_Br and F_2_SO, respectively; for these, mapping on the 0.0020 a.u. isodensity envelope provided a description of the strength and the complete nature of the electrostatic potential (and hence on the reactivity) for the entire set of monomers examined;The σ-holes on the fluorine atoms in most of the monomers in the series are positive. The positive σ-holes were capable of making attractive engagements with the positive σ-hole in the interacting molecule(s) in governing σ···σ stabilizations, leading to the formation of the dimers;A negative σ-hole does exist on covalently bound fluorine atoms in molecules; this is shown for ClF_2_Br, ClF_3_, and SOF_2_ on the Cl–F, Cl–F and S–F bond extensions;For all the dimers examined, the appearance of an intermolecular bond-path and critical-point topology in the intermolecular regions is consistent with what was inferred from energetics and intermolecular distances; thus, their use is always recommended to better understand non-covalent interactions in both simple and more complex situations;Utilization of the MESP-only approach led to ambiguous conclusions on the ability of a specific positive site on a monomer to be involved in an attractive engagement with a positive site on another monomer, answering the question posed in the title of this paper;The SAPT approach is useful in instances where a simple bond path topology, molecular electrostatic potential, and the binding energy are insufficient to explain the origin of a perhaps counterintuitive intermolecular interaction. For all the fluorine-centered interactions examined, the dispersion term dominates over the electrostatic term; accordingly, these interactions could be regarded as dispersion driven. However, the electrostatic interaction, together with other interacting contributions, was also shown to play a role that cannot be overlooked;The X···F (X=F, Cl, Br) interactions discussed in this study display a *σ-**σ centered type-II halogen bonding* pattern that is very rarely discussed in the non-covalent chemistry literature, which is certainly not halogen bond.

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
