# Peer review of "Is the Fluorine in Molecules Dispersive? Is Molecular Electrostatic Potential a Valid Property to Explore Fluorine-Centered Non-Covalent Interactions?"

_molecules, 2019, doi:10.3390/molecules24030379_

Round 1

Reviewer 1 Report

The manuscript is very interesting indeed but I have some comments that should be addressed. In general, the tone of the article is quite aggressive and the authors try to reduce a very complex problem to one single variable. Besides, I cannot see any novelty since what the authors claims, which F···F is not electrostatic, it is already known.

First of all, the manuscript is difficult to read, there are so many “unsolved” questions and it is easy to lose the focus on what the authors what to do. Please reduce redundant sentences and question can be easily moved into the results section so they can be clarify using the pertinent results

Too many questions made the reader lose the focus of the article. It looks like an infinite of unsolved questions. It will be better to merge some of the questions and avoid repetitions.

“ 0.001 a.u. isodensity envelope for mapping the electrostatic potential is found to provide incorrect information about the nature of the surface reactive sites on some of the isolated 29 monomers, which can mislead a misinterpretation of the results obtained.”

0.001 au isodensity resembles to the vdW. It is quite established that this is a good option to map the MEP but also 0.002

130 If it does, is the electrostatic energy that arises from Coulombic interaction between the interacting

 species sufficient to explain the attraction between the two positive sites? It is known that F···F interaction are not pure Coulombic but driven by dispersion forces.

158: “The energy minima found through geometry relaxation may be shallow, but is cannot be presumed that they have no real impact on crystal structure formation. Such a view is reminiscent of that held years ago that fluorine does not halogen bond, and now proven incorrect.” That’s a finding or a result from their calculations but it is in the introduction, so it shouldn’t be here.

Scheme 1 is not completed. Image seems to be cropped on the left hand side.

Line:189. Mixing basis set is not a well practice in order to compare results. I would suggest to stick to the same basis set for all the optimisations and energies comparison. If Dunning’s basis set is too large and computationally speaking unaffordable, authors should do all the calculations with Pople’s, including CCSD(T).

Why the MEPs on the vdW were obtained using M06-2X when MP2 calculations are available? Could the authors show also MP2 MEP results?

QTAM, what wavefunction was used to do this calculations? M06’s or MP2’s?

SAPT, again, there is a mixture of basis sets, now authors used jul-cc-pVDZ for SAPT (no reasons given though), CCSD(t) with aug-cc-pVTZ and optimisations with M06/MP2 and Pople’s. If not strong reason is given, all the calculations should be done within the same basis set.

And, all the unit should be the same as well. Some authors (in which I am myself) consider that MEP should be given in a.u. while binding energies in kJmol-1.

Line 211:  MESP model. This should be defined.

Line 212: There is precedence that the VS,max on the halogen in an isolated monomer correlates with the binding energy of the halogen bonded complex, and can thus be regarded as a measure of bond strength.” This is not only a precedent but also proven by Polizter et al, Frontera et al, etc. When a pure electrostatic interaction through the s-hole is done, there is a linear correlation between the depth of the sigma hole and the binding energy. But the opposite is not true, no correlation between both quantities does not imply that the interaction is not doe through the s-hole

Line214: While doing so, the importance of the electron density changes, which is expected when two systems are in the close proximity, associated with the interacting atoms upon the formation of the non-covalent interactions is neglected. “ I don’t know what the authors mean here. Please clarify

Line 217: This is because VS,max cannot be calculated on the bonding region since the atoms that are 217 bonded non-covalently have already (at least partially) overlapped at equilibrium.” This is very obvious, since the molecules are already interacting their vdW surfaces are , by definition, overlapping, i.e. MEP on the vdW is within already interacting molecules is useless.  I can’t see the point behind this statement. Again, MEP is beyond any doubt obtained and used to explain nucleophilic/electrophilic attacks. No point to reinforce this idea, while it a level-zero knowledge on non-covalent interactions field.

Figure 1. Colour coding range should be provided with values and all the molecules should be plotted within the same range.

238: Since s-hole regions on these halogen atoms in the chosen molecules are positive, they should not attract the positive site on another species;.  if they do so twould be counter to the fundamental principle of Coulomb’s law.” This is a very simplistic view of a very complex problem. It’s well-known that F···F interactions happens and are mainly due to dispersion forces. The whole article relies on the contradiction between the F····F interaction and the existence of positive s-holes. Even interactions between two anions (ref) are possible.

240: Now the question arises: are the predicted 0.0010 a.u. isodensity mapped surface maxima of electrostatic potential Vs,max (s-hole strengths) on the halogen atoms in monomers correct? They may be correct as long as the strengths of the positive -holes on the halogen surfaces are reasonably strong.” I disagree with this statement. What is the correctness of the MEP max? It’s known that the mapping of the MEP on the vdW implies a approximation of the electron density to a isosurface (usually done with -sign of the volume of the tetrahedron formed between 4 point in the molecule). This implies that some s-hole can be obscured by the grid definition, since they are too “small”. So, the fact that a Vs,max is not obtained on the vdW isosurface, doesn’t (always) implies that it doesn’t exist. Also, complicated electron density isosurfaces can lead to s-hole hidden inside or partially obfuscated. Again, I feel that the authors try to reduce a complex problem to a single variable.

267: These results suggest that the 0.0010 a.u. isodensity surface is not always appropriate for the prediction of the nature of electrostatic potential on the surface of a chemical system. This means the choice of an appropriate isodensity envelope has a significant impact in revealing the actual nature of the reactive profile of an atom in molecules.” I don’t think that this statement is correct. The election of the electron density isosurface is quite arbitrary and it is not written on stone. One can decide that 0.0015 is better option. The limit of the molecular space is quite difficult to predict since it depends on the type of atom considered. For example, hard and compact atoms like F will be more suitable for a larger value of the electron density isosurface since the density is more concentrated due to the electronegativity, and smaller values ( such as 0.001au) could be less sensitive. But this doesn’t mean that the whole 0.001 isosurface is not appropriate for prediction of the reactivity of the whole molecule. This is again narrowing down a complex problem to a single and arbitrary variable. As authors stated few lines after, this is simply to check with another value. But from their values of Table 1, what I can see is that both isovalues are quite consistent with few exceptions.

Figure 2 is also cropped. Also, I would like to highlight that it may happen that M06-2X is not the best option to analyse those complexes. Alkorta et al analysed XNO2 homodimers and heterodimers at the MP2/aug-cc-pVTZ computational level and found the F···F distance in FNO2 homodimer was 2.805 rather than 2.775A as predicted by the authors (New J. Chem., 2015,39, 6791-6802 , and New J. Chem., 2016,40, 9060-9072 )

Line 329: Scheme 2 should be modified to include Type-I complexes.

Line 412: The correlation between the electron density at the BCP and the strength of the bond is questionable. In fact, it has been shown that the absence of the BCP doesn’t imply the absence of interaction.

The role of the dispersion provided by SAPT in the F···F interaction is not new nor surprising. It was already pointed out but Alkorta and Quiñonero win XNO2 complexes, including FNO2 (New J. Chem., 2015,39, 6791-6802 , and New J. Chem., 2016,40, 9060-9072)

Author Response

(Attached)

Reviewer 2 Report

I recently reviewed this manuscript for J Phys Chem. It was obviously rejected. As far as I can detect, the authors have simply resubmitted the manuscript to Molecules. I regard this behavior as unethical as I provided a detailed critique of the scientific deficiencies of the work that has been ignored completely. 

My original report (also unchanged) is attached 

Author Response

(attached)

Reviewer 3 Report

Comments on the manuscript ID molecules-413824:

Title: Is the Fluorine in Molecules Dispersive? Is Molecular Electrostatic Potential a Valid Property to Explore Fluorine-Centered Non-Covalent Interactions?

Authors: Arpita Varadwa, Helder. M. Marques, Pradeep R. Varadwaj

   This work reports on some properties of F—F bonds discussing relevant features of the halogen-halogen interactions. Despite being very electronegative species, the authors state that the interaction between them can play an important role in structural matters. This is an interesting topic which has attracted renewed attention in the last years and there are still many open questions. After analyzing several molecules containing fluorine and the corresponding dimers they conclude that the dispersion term contribution is essential to achieve a proper description of the interaction.

   As I have mentioned the topic is interesting and the calculations are comprehensive and appropriate. Although this paper is not going to solve the problem of the halogen-halogen bond description in a definitive way, it provides useful information that could be taken into account in further studies, especially those containing experimental evidences. Thus, I would recommend the publication of this work if some specific comments are considered.

1)      Fig. 1 does not have a colorbar showing, at least the relative variations of the potential. This is an essential point to give in the figure in order to provide a qualitative description of the molecules referred. In addition, the explanations about this issue in the maintext are also somehow unclear. Further rewriting work should be necessary to achieve a better understanding of this part. If the relative variations of the molecules are different in each case, this fact could be the responsible for this obscure description.

2)      The method section does not provide important details of the calculations. For example, the algorithm used for the optimization of the geometries, the starting configurations of the dimers, if they are some atoms fixed or not, etc. Besides, they use several computational methods in this work. The authors must clarify in the method section why is used each method and for which purpose. This information at that point of the text should be very useful for the reader before looking at the results.

3)       Finally, this paper lacks of a true conclusions section. This is an important part, especially for such a long paper where a lot of information is provided, and surely not all the results are comparable in importance. Therefore, the authors should mandatorily include a conclusions section (rather than a summary) in which it was completely clear which results are new in this work and which are the main results that have not been reported so far.

Author Response

(Attached)

Reviewer 4 Report

In this manuscript, the authored state-of-the-art computational chemistry approaches to understand intermolecular interactions involving substituted halogen (particularly fluorine) atoms. The authors studied the electrostatic potential surface of 17 halogenated molecules, and the intermolecular interactions of 22 binary complexes made of those halogenated molecules. They further analyzed the non-covalent intermolecular interactions using Quantum Theory of Atoms in Molecules (QTAIM) and Symmetry Adapted Perturbation Theories (SAPT). The computational chemistry analysis is comprehensive and sufficient data are provided to support their arguments. The results are of interests to people who are studying halogen chemistry. I would recommend the manuscript to be published in the journal of Molecules subjected to some minor revisions. 

1. In Scheme 1, structures k and q are either incomplete or incorrect. Please correct them. 

2. In Figure 2, structures I and v are incomplete. Please correct them 

3. In Table 3, imaginary frequency exists in some complexes such as F2…Cl2. Are these complex structures stable in reality? Please explain that in the main text. 

4. Please remove ‘on’ in Line 485, on Page 14 of 25. 

Author Response

(Attached)

Round 2

Reviewer 1 Report

I can see that the authors have indeed done a great effort to answer all my questions and the manuscript has been considerably improved. There are still some questions to be answered and I am not completely in agreement with some statements, but I understand that this is a long-term discussion and honestly I found this paper quite stimulating. I would suggest the acceptance of this manuscript. Only two things:

Reply: We thank the reviewer for comments. We have tried to improve the tone of the paper. That “F...F is not electrostatic” is actually not always true, and this is shown in this paper.

Re-Reply: I am in disagreement with this. Authors clearly show in Table 2 that for all the interactions observed, the dispersion term governs over the electrostatic one, so according to SAPT, those interactions are mainly driven by dispersion forces. So, their statement in line 615: “This shows that the F···F interactions in these 14 complexes are partially stabilized by electrostatics” is not entirely correct and misleading. Electrostatic always plays some role in the interaction, but the leading one (in this case) is dispersion. The line 615 should be removed since it is misleading.

The role of the dispersion provided by SAPT in the F...F interaction is not new nor surprising. It was already pointed out but Alkorta and Quinonero win XNO2 complexes, including FNO2 (New J. Chem.,2015,39, 6791-6802 , and New J. Chem., 2016,40, 9060-9072)

Reply: These papers of Alkorta et al are cited.

Re-Reply: I couldn’t find those citations in the new version of the manuscript.

Author Response

(Attached)

Reviewer 2 Report

Please see the attached PDF

Author Response

(Attached)

Reviewer 3 Report

Comments on the manuscript ID molecules-413824 (round2):

Title: Is the Fluorine in Molecules Dispersive? Is Molecular Electrostatic Potential a Valid Property to Explore Fluorine-Centered Non-Covalent Interactions?

Authors: Arpita Varadwa, Helder. M. Marques, Pradeep R. Varadwaj 

   Some changes have been introduced in the main text in this revision version of the manuscript. However, some of the problems that I pointed out in the first round of comments remain unattended. The method section is still obscure and they do not provide important details regarding important aspects essential to reproduce these calculations. “For example, the algorithm used for the optimization of the geometries, the starting configurations of the dimers, if they are some atoms fixed or not, etc. Besides, they use several computational methods in this work.” This paragraph is copied from my first report.

   In addition, they have included some conclusions of the work, but I am afraid that these statements are not deduced exclusively from the results obtained in this work. These difficulties to distinguish between the really new results and other previous findings from the literature is due the excessive length of manuscript, in which they appeared mixed results from the present work, excerpts and citations from other reports in the bibliography or simply speculative explanations.  It is true that this is a topic extensively treated in the literature and it makes hard to write a good manuscript taking into account the previous work and incorporating the new results. However, the clarity in this regard must be mandatorily very high.  Otherwise, it is impossible to discern the real contributions of this work from other previous results.

   Thus, attending to these considerations I am not able to further recommend the publication of this work. My recommendation is to carry out a complete rewriting of the text before an eventual resubmission. The new aim of the text should be focused exclusively on the new results (as expected in a journal paper) and leaving apart the discussions regarding other views of the problem, that would be more appropriate for a book chapter or other kind of publication.

Author Response

(Attached)
